# Rel Family Transcription Factor NFAT5 Upregulates COX2 via HIF-1α Activity in Ishikawa and HEC1a Cells

**DOI:** 10.3390/ijms25073666

**Published:** 2024-03-25

**Authors:** Toshiyuki Okumura, Janet P. Raja Xavier, Jana Pasternak, Zhiqi Yang, Cao Hang, Bakhtiyor Nosirov, Yogesh Singh, Jakob Admard, Sara Y. Brucker, Stefan Kommoss, Satoru Takeda, Annette Staebler, Florian Lang, Madhuri S. Salker

**Affiliations:** 1Department of Women’s Health, Tübingen University Hospital, D-72076 Tübingen, Germany; tokumura@juntendo.ac.jp (T.O.); janet.raja-xavier@med.uni-tuebingen.de (J.P.R.X.); jana.pasternak@med.uni-tuebingen.de (J.P.); caohang3@gmail.com (C.H.); yogesh.singh@med.uni-tuebingen.de (Y.S.); sara.brucker@med.uni-tuebingen.de (S.Y.B.); stefan.kommoss@diakoneo.de (S.K.); 2Department of Obstetrics and Gynecology, Faculty of Medicine, Juntendo University, Tokyo 113-8421, Japan; stakeda@juntendo.ac.jp; 3Department of Cancer Research, Luxembourg Institute of Health, L-1210 Luxembourg, Luxembourg; 4Institute of Medical Genetics and Applied Genomics, Eberhard Karls University, D-72074 Tübingen, Germany; jakob.admard@med.uni-tuebingen.de; 5Institute of Pathology, Eberhard Karls University, D-72074 Tübingen, Germany; annette.staebler@med.uni-tuebingen.de; 6Institute of Physiology, Eberhard Karls University, D-72074 Tübingen, Germany; florian.lang@uni-tuebingen.de

**Keywords:** endometrial cancer, hypoxia inducible factor 1α, cyclooxygenase 2

## Abstract

Nuclear factor of activated T cells 5 (NFAT5) and cyclooxygenase 2 (COX2; *PTGS2*) both participate in diverse pathologies including cancer progression. However, the biological role of the NFAT5-COX2 signaling pathway in human endometrial cancer has remained elusive. The present study explored whether NFAT5 is expressed in endometrial tumors and if NFAT5 participates in cancer progression. To gain insights into the underlying mechanisms, NFAT5 protein abundance in endometrial cancer tissue was visualized by immunohistochemistry and endometrial cancer cells (Ishikawa and HEC1a) were transfected with NFAT5 or with an empty plasmid. As a result, NFAT5 expression is more abundant in high-grade than in low-grade endometrial cancer tissue. RNA sequencing analysis of NFAT5 overexpression in Ishikawa cells upregulated 37 genes and downregulated 20 genes. Genes affected included cyclooxygenase 2 and hypoxia inducible factor 1α (*HIF1A*). NFAT5 transfection and/or treatment with HIF-1α stabilizer exerted a strong stimulating effect on HIF-1α promoter activity as well as COX2 expression level and prostaglandin E2 receptor (PGE2) levels. Our findings suggest that activation of NFAT5—HIF-1α—COX2 axis could promote endometrial cancer progression.

## 1. Introduction

Endometrial cancer (EnCa) is a frequent gynecological neoplasia and its incidence rate has increased in the past years, especially in developed or high-income countries [1,2]. In 2020, the reported global incidence of EnCa was 417,367 and about 2.2% of newly diagnosed cancer cases in that year, making it the sixth most commonly diagnosed cancer [3]. In contrast to breast and cervical cancers, which are expected to decline significantly by 2030, EnCa is expected to increase over the next decade [2,4,5]. The largest incident rate has been reported in women aged 65–79 years [6]; however, alarmingly, EnCa is increasing among younger age groups (25–49 years and 50–59 years) as well [7,8]. Risk factors for EnCa include ethnicity (in particular Caucasians), poor diet (high salt/sugar) obesity, nulliparity, polycystic ovarian syndrome, estrogen-only hormone replacement therapy (HRT), comorbidities such as hypertension or diabetes, and genetic predisposition (Lynch and Cowden syndromes) [9,10,11]. As the world’s population increases, together with an ageing population and the prevalence of risk factors, the number of individuals with EnCa is expected to increase further.

EnCa is a hormone-sensitive disease thought to arise due to excessive estrogenic stimulation of the endometrial lining of the uterus [4], though estrogen-independent pathways are also known to participate in carcinogenesis. This aberrant stimulation leads to mitogenic activation by hijacking physiological cellular mechanisms (i.e., the down regulation of checkpoint mechanisms, thus leading to uncontrolled proliferation), ultimately leading to malignant transformation and metastasis [11]. Cancer can advance when it acquires six biological hallmarks of cancer development, including: persistent proliferation signaling, evasion of growth suppressors, resistance to cell death, immortality, induction of angiogenesis, and activation of invasion and migratory pathways [11,12]. The molecular mechanisms behind these events involve a complex interplay of genetic, epigenetic, and environmental factors that lead to the uncontrolled growth and division of cells [9,13]. The survival outcomes are poor in patients with advanced disease, and hence, there is an urgent need for identification and discovery of new molecular targets to improve the survival of patients with EnCa.

Nuclear factor of activated T cells 5 (NFAT5/TonEBP) is a member of the Rel family of transcriptional activators, which also includes nuclear factor κappa B (NFκB) and NFAT1–4. NFAT5 has a DNA binding domain that is in sequence homology with the rel homology domain (RHD) [14]. In contrast to its other isoforms (NFAT1–4), NFAT5 lacks a calcineurin-binding domain outside of its DNA binding domain [14]. The NFAT5 protein contains a leucine-rich nuclear export sequence (NES) followed by a proline-rich transactivation domain (AD1) at the N-terminal. Further, it has low-complexity glutamine and serine/threonine-rich regions (AD2 and AD3) at the C- terminal end. The three activation domains (AD1, AD2, and AD3) act in coordination in response to hypertonicity [15,16]. NFAT5 is expressed in tissues that are often subjected to osmotic stress, such as the renal medulla, eyes, and skin [17,18,19]. Separately from the well-known osmoprotective role of NFAT5 in the renal medulla, NFAT5 is also activated in a *tonicity-independent* manner, having broader implications in development, immune function, and cellular stress responses [14]. Aberrant NFAT5 levels contribute to several pathologies, including hypoxia [20], vascular calcification [16,21], diabetes [22], inflammation [15,23], chronic kidney disease [24], bacterial infection [25], and are seen in breast and lung cancer [26,27,28]. NFAT5 is also highly expressed in mouse, ovine, and human placenta and throughout gestation in the mouse embryo, suggesting its critical role during embryonic development and fetal maturation [29]. NFAT5’s involvement in cancer pathogenesis is not as extensively studied as other transcription factors; however, there is evidence suggesting its potential impact on tumor development. Studies have explored its involvement in breast cancer, renal cell carcinoma, and glioblastoma [30,31,32]. NFAT5 has been implicated in promoting cell survival and proliferation in breast cancer cells via secretion of pro-angiogenic factors [30]. High expression of NFAT5 levels in renal cell carcinoma has been correlated with various clinicopathological features, including tumor stage, grade, and metastasis [31]. This suggests NFAT5’s potential role in the aggressiveness and progression of tumor pathophysiology. These findings suggest a potential role for NFAT5 in cancer pathogenesis; however, it has not yet been described in EnCa.

Hypoxia (insufficient oxygen) is a common feature of most tumors [33,34]. Hypoxia-inducible factors (HIFs) are often upregulated in cancer microenvironments and are known to aggravate tumorigenesis by inducing epithelial–to–mesenchymal transition (EMT) and induce a stem-cell-like phenotype, thus promoting cell survival [34,35]. A hypoxic environment can exacerbate tumorigenesis but is also involved in reducing therapeutic efficacy of chemotherapeutic agents [17,36]. Thus, identifying new factors inducing and driving hypoxia signaling in cancer progression is important in the development of new therapeutic targets for EnCa. Several studies show that cellular activation of NFAT5 favors transcriptional activation of HIF-1α [20,25,37]. NFAT5 and HIF-1α are known to coordinate together [25]. As highlighted by Yang et al., coordinated functional regulation between NFAT5 and HIF-1α is critically important in the pathogenesis of hypoxic-ischemic encephalopathy [20]. The cooperation of NFAT5 and HIF-1α in the pathophysiology of EnCa has not yet been defined. 

In kidney and colon cancer models, NFAT5 has been shown to upregulate cyclooxygenase 2 (COX2; *PTGS2*) [38,39], an enzyme of paramount functional importance in both normal and malignant endometrial tissue [40,41]. Strikingly, inflammation-associated COX2 activation is also known to enhance HIF-1α activity in some tumor models such as breast and lung cancer [42,43]. HIF-1α-dependent COX2 activation is shown to promote proliferation, inflammation, and tumor metastasis [44]. It is currently unknown if NFAT5 is present in endometrium and if there is a putative link to NFAT5 activation and HIF-1α/COX2 signaling axis in EnCa progression.

The present study thus explored whether NFAT5 is expressed in human endometrial cancer tissue, whether NFAT5 expression in endometrial cancer cells is sensitive to HIF-1α, and investigated whether NFAT5 activation influences the expression of HIF-1α and COX2 signaling cascade in EnCa pathogenesis. 

## 2. Results

The present study explored the expression and function of the transcription factor NFAT5 in endometrial cancer. A total of 26 cases were selected at random (between 2014–2016, Table 1). From the cohort, *n* = 15 were over the age of 60 and the vast majority of the cases had a post-menopausal status. Of the cases investigated the *n* = 24 were classified as Endometriod and two were Serous histotype.

The overall score of the staining intensity typically has the following categories: weak (score 1), moderate (score 2), and strong (score 3) [45]. Immunostaining was performed on formalin fixed, paraffin-embedded (FFPE) archival endometrial tumor tissue and investigated for NFAT5 expression. As apparent from immunohistochemistry (Figure 1a and Table 1) in low-grade (G1, G2) endometrial cancer tissue, NFAT5 staining showed low to intermediate cytoplasmic intensity (score 1) in the tumor cells, which is less than the moderate staining in neighboring endothelial cells (score 2) serving as an internal reference. In contrast, NFAT5 expression shows a strong and block-like expression pattern in high-grade endometrioid carcinomas (G3) with particularly strong staining in the perivascular area and on the leading edge. The staining intensity is clearly homogenous and stronger (score 3) than in the adjoining endothelial cells (moderate intensity, score 2). Benign endometrium showed intense expression (score 3) in proliferating glands and reduced or low expression (score 1) in non-proliferating cells in the secretory phase. We observed a significant association between a higher grade and intense NFAT5 staining (*p* < 0.001). Additionally, with cases diagnosed with pT1b and higher (i.e., invasion into the outer half of the myometrium), we noticed a significant association with an increase of NFAT5 staining (*p* = 0.014). An increase with NFAT5 staining is also significantly associated with metastasis (*p* = 0.043). The staining pattern of NFAT5 was similar in other G1 and the G3 cases verified in this study (Appendix A). 

In parallel, total RNA was collected from the same FFPE blocks and subjected to qRT-PCR. RNA data revealed that *NFAT5* transcript levels were also significantly higher in G3 (aggressive) tumor tissues (Figure 1b, *, *p* = 0.0130) compared to low-grade G1 tumor tissues. 

To gain further insight into the role of NFAT5 overexpression in high-grade endometrial cancer, we used Ishikawa cells, a well-used model of adenocarcinoma cancer [46]. Total RNA was harvested from four independent cultures, following transfection with either NFAT5 overexpression plasmid or an empty vector for 24 h. 

After accounting for variations between cultures, the effect of NFAT5 overexpression on the gene expression pattern in Ishikawa cells was observed to be highly consistent. Based on FDR < 0.05 and log2FC ≥ 0.3 (corresponding to actual fold change ≥ 1.23), we identified 57 differently regulated genes that were significantly altered upon NFAT5 overexpression in Ishikawa cells (Figure 2a), of which 37 genes were upregulated and 20 genes were downregulated. Genes upregulated significantly upon NFAT5 overexpression in Ishikawa cells include leucine-rich repeat containing G protein-coupled receptor 6 (*LGR6*) (log2FC = 1.749), *NFAT5* (log2FC = 2.705), prostaglandin-endoperoxide synthase 2 (*PTGS2*, which encodes for COX2 protein) (log2FC = 0.320), netrin 4 (*NTN4*) (log2FC = 0.476), and angiogenin (*ANG*) (log2FC = 0.566). NFAT5 overexpression in Ishikawa downregulated genes like ankyrin repeat domain 1 (*ANKRD1*) (log2FC = −0.855) coding, a transcription factor that positively regulates apoptosis [47] and amine oxidase copper containing 3 (*AOC3*) (log2FC = −0.358), which at low levels is associated with poor prognosis in cancers [48]. Although *HIF1A* (log2FC = 0.104, FDR < 0.05) and estrogen receptor 1 (*ESR1*) (log2FC = −0.22, FDR < 0.05) were not differentially expressed based on the differential expression thresholds we used in this study, we observed modest but significant changes in their expressions too upon NFAT5 overexpression. The log2 fold change values of differential gene expression between the control and NFAT5 overexpression in Ishikawa, for genes of interest, are represented in Figure 2b. Taken together, these results show there is an increase of *PTGS2* and *HIF1A* transcripts after overexpression of NFAT5 in Ishikawa. The complete set of differently expressed genes are shown in Appendix A. In order to examine the role of molecules predicted to be involved in the pathways relevant for cancer progression, we used Ingenuity Pathway Analysis (IPA), a bioinformatics tool from QIAGEN, to examine the underlying molecular mechanisms. Appendix A points to a strong association with activation of PTGS2 (COX2) signaling upon NFAT5 overexpression in Ishikawa cells. 

To test whether induction of NFAT5 expression in Ishikawa cells is sensitive to hypoxia, cells were treated with a known cell permeable prolyl-4-hydroxylase (PHD) inhibitor [49], dimethyloxalylglycine (DMOG) (0.5 mM DMOG for 24 h). PHD is involved in degradation of HIF-1α and during hypoxia, PHD is blocked due to limited oxygen, leading to HIF-1α stabilization and an increase in downstream target gene activation [49]. DMOG is known to suppress PHD activity and stabilize HIF-1α levels, thus maintaining hypoxic environment both in vitro and in vivo conditions [49,50,51]. Ishikawa was treated with DMOG for 24 h and total RNA was extracted. qRT-PCR was performed for NFAT5 gene expression analysis. As shown in Figure 3a (*n* = 6; *, *p* = 0.0107), DMOG treatment was indeed followed by a significant increase of *NFAT5* transcript levels. Western blotting was employed to test whether the stimulation of NFAT5 transcription was followed by an increase of protein expression. As illustrated in Figure 3b and Appendix A (*n* = 6; ****, *p* < 0.0001), the effect of DMOG on *NFAT5* transcript levels was paralleled by a highly significant increase in NFAT5 protein abundance in Ishikawa cells.

To gain insight into whether enhanced expression of NFAT5 can induce the HIF-1α signaling axis, Ishikawa cells were first transfected with either NFAT5 overexpression plasmid or an empty vector for 24 h and then subjected to qRT-PCR. As illustrated in Figure 3c (*n* = 5; **, *p* = 0.002, **, *p* = 0.005, ****, *p* < 0.0001), NFAT5 transfection was followed by the expected up-regulation of *NFAT5* transcript levels and by a significant increase of *HIF1A* and *PTGS2* transcript levels. As illustrated in Figure 3d and Appendix A (*n* = 6; *, *p* = 0.037; **, *p* = 0.008), the effect of NFAT5 transfection on *NFAT5* and *PTGS2* transcript levels was paralleled by a similar increase of NFAT5 and COX2 protein abundance. As shown in Figure 3e (*n* = 6; ****, *p* < 0.0001), NFAT5 overexpression significantly increased HIF-1α activity, as measured by hypoxia response elements (HRE)-luciferase. These data suggest that NFAT5 can induce *HIF1A* transcription and activity in Ishikawa cells. Further, HIF-1α activation can increase COX2 levels in colon cancer cells [43,52]. To test this hypothesis in endometrial carcinoma cells, the next series of experiments tested whether COX2 expression is sensitive to DMOG (i.e., hypoxia). As illustrated in Figure 3f (*n* = 6; **, *p* = 0.001), DMOG treatment in Ishikawa cells was followed by a significant increase of *PTGS2* transcript levels. Following an increase of *PTGS2* gene expression, we observed an increase of COX2 protein abundance (Figure 3g and Appendix A, *n* = 6; **, *p* = 0.001) upon DMOG treatment in Ishikawa cells. 

NFAT5 has been shown to upregulate *PTGS2* gene expression (COX2) [33,34]. In turn, COX2 activation is also known to enhance HIF-1α activity in breast and lung cancer [37,38]. To address whether NFAT5 overexpression can lead to a more aggressive state, we monitored cell cycle progression, cell proliferation, and cell migration in Ishikawa cells transfected with NFAT5 plasmid. The examination of the cell cycle profile (Figure 4a, *n* = 7; *, *p* = 0.012) revealed a higher proportion of cells in the S phase upon NFAT5 overexpression, indicating an augmentation in DNA replication compared to the control. Cells present at G0/G1 phase of the cycle showed no difference between the control and NFAT5-transfected population. Further, we observed fewer numbers of NFAT5-transfected cells at the G2/M phase compared to the control cells. As shown in Figure 4b (*n* = 4; **, *p* = 0.002), the overexpression of NFAT5 resulted in a significant increase in cell proliferation as established with BrdU ELISA. Furthermore, the wound healing assay demonstrated a significant enhancement in cell migration in Ishikawa cells at 24 h post scratch following NFAT5 overexpression (Figure 4c,d, *n* = 4; **, *p* = 0.002).

To test if there is positive feedback between NFAT5 and hypoxia, we transfected Ishikawa cells with NFAT5 overexpression plasmid followed by DMOG treatment or DMOG alone. As a result (Figure 5a, *n* = 6; *, *p* = 0.0117; **, *p* = 0.0064), we observed significantly higher *NFAT5* and *PTGS2* transcript levels compared with only DMOG treatment. As illustrated in Figure 5b–d and Appendix A (*n* = 5; *, *p* < 0.05; **, *p* < 0.01; ***, *p* < 0.001, ****, *p* < 0.0001), the effect of NFAT5 transfection and DMOG treatment was paralleled by a similar increase of NFAT5 and COX2 protein abundance compared to cells treated with DMOG alone. To explore the putative link with NFAT5 and hypoxia, HIF-1α promoter activity was thus quantified by dual-luciferase reporter assays with prior 24 h NFAT5 transfection and/or treatment with DMOG. As illustrated in Figure 5e (*n* = 6; **, *p* = 0.005), NFAT5 transfection and DMOG treatment in Ishikawa cells both exerted a strong stimulating effect on HIF-1α promoter activity, which was further increased by combined NFAT5 transfection and DMOG treatment. These results were also paralleled by an increase in secreted PGE_2_ levels in Ishikawa cells with a significant difference observed in cells with combined NFAT5 transfection and DMOG treatment Figure 5f (*n* = 4; *, *p* = 0.024, *p* = 0.012). 

In order to test whether NFAT5 overexpression is sensitive to hypoxia in an alternative endometrial cell line, HEC1a cells were initially transfected with either an NFAT5 overexpression plasmid or an empty vector for a duration of 24 h. Subsequently, the cells were treated with or without DMOG at a concentration of 0.5 mM for an additional 24 h. Appendix A demonstrates that the NFAT5 overexpression followed by DMOG treatment did induce an increase in NFAT5 protein expression higher than that following only DMOG treatment in HEC1a cells (*n* = 3, *, *p* = 0.014). Furthermore, analysis of HIF-1α promoter activity showed a significant elevation upon NFAT5 overexpression and a further increase upon DMOG treatment (Appendix A, *n* = 4, ***, *p* < 0.001). Investigation on HEC1a migratory potential upon NFAT5 overexpression revealed an increase in cell migration at 24 h with the wound scratch assay (Appendix A). Taken together, these results confirm NFAT5 responsive hypoxia induction in HEC1a, verifying NFAT5 relevance in other adenocarcinoma cell lines. 

## 3. Discussion

There has been an astonishing increase of reports describing the role of NFAT5 in tonicity-independent manner with its effects in cell development and human diseases [27,28,53,54,55]. NFAT5 is a pleiotropic stress protein with a protective role in cellular adaption to osmotic stress, bacterial infection, and genotoxin-induced DNA damage [18,56]. NFAT5-mediated pathological responses can result in human pathologies such as autoimmune diseases, acute kidney injury, hepatocellular carcinoma, atherosclerosis, and obesity [15,56,57,58]. Importantly, downregulation of NFAT5 reduces inflammation and thereby renders a protective role in preventing these diseases [15]. NFAT5, as a protective factor, is well studied; however, its role in endometrium and tumor progression and metastasis is still in its infancy. 

In our study, we report that expression of NFAT5 was significantly higher in more aggressive (G3) endometrial cancer tissues than in corresponding non-tumor, low-grade (G1) tissues. Staining was highest around blood vessels and the leading edge. Our study was using a small proof-of-concept cohort and further larger clinical cohorts are required to validate our findings. To gain further insight of NFAT5 overexpression and establish a comprehensive analysis of aberrantly expressed genes after NFAT5 overexpression in Ishikawa cells (well used model cells of endometrial adenocarcinoma [59]), RNA-sequencing was performed. NFAT5 plasmid transfection into Ishikawa cells upregulated 37 genes and downregulated 20 genes. In keeping with its established role, many of the NFAT5-regulated genes were related to osmoregulation and/or support cell survival in hypertonic environment (e.g., aldo-keto reductase family 1 member b (*AKR1B1*), solute carrier family 6 member 12 (*SLC6A12*), and ATPase Na^+^/K^+^ transporting subunit beta 1 (*ATP1B1*) [19,60]. Interestingly, altered genes after NFAT5 overexpression in Ishikawa cells included *PTGS2* and *HIF1A*. Given the hypoxia-sensitivity of NFAT5 expression, we posit that local hypoxia of advanced cancer tissue contributes to or even accounts for upregulation of NFAT5 expression in EnCa [25,53,61].

It has been established that HIF-1α regulates oxygen homeostasis in the tumor microenvironment and can elevate COX2 expression by regulating it at transcriptional level [52,62]. Further, it has also been linked with increased levels of PGE_2_ and cancer progression [52]. These results are in concordance with our findings reported here. Along this line, NFAT5 transfection in Ishikawa cells exerted a strong stimulating effect on HIF-1α promoter activity. Furthermore, the HIF-1α-stabilizing prolylhydroxylase inhibitor, DMOG, significantly increased *NFAT5* transcript levels and protein abundance in Ishikawa cells. Thus, our results show that NFAT5 upregulates *HIF1A* transcription and promoter activity in Ishikawa cells. Further, the stimulation of HIF-1α via NFAT5 with DMOG augments the response, and thus establishes a positive feedback loop.

COX2 is a well-known mediator of pro-tumorigenic inflammation. It is upregulated in many cancers and is involved in tumor progression [63,64]. COX2 is responsible for the synthesis of prostanoids (prostaglandins, prostacyclin, and thromboxane) from the precursor arachidonic acid [65]. Prostaglandins trigger the release of proinflammatory chemokines [66,67]. Indeed, it is now well established that inflammation is a critical and enabling characteristic of tumorigenesis [68]. Our results show an increased trend of PGE_2_ levels upon NFAT5-COX2 signaling activation in Ishikawa cells. The ELISA approach employed in this study holds some limitations such as its inability to detect low levels of secreted PGE_2_ or that considering its rapid half-life, it is not able to detect PGE_2_ quickly [69]. Therefore, we propose the future utility of liquid-chromatograph-based mass spectroscopy (LC-MS/MS) approaches evaluating these lipid molecules. Meanwhile, it is also worthy to analyze if other isoforms of prostanoids are regulated upon NFAT5-mediated COX2 activation.

This synergism towards tumor progression and metastasis makes COX2 a potential therapeutic target. Since chemoresistance is very closely associated with hypoxia and COX2 overexpression in tumor, inhibiting of COX2 activity may result in increased effectiveness of cancer therapies (chemotherapy and radiation) [70]. Furthermore, selective inhibition of COX2 with nimesulide was successful in able to reducing tumor formation in a mouse hypoxic tumor model. Further work is warranted to verify if this can be confirmed in humans [71].

We show that NFAT5 transfection and DMOG treatment were followed by significant increases of transcript levels and protein abundance of COX2, another signaling molecule sensitive to hypoxia [52,72]. NFAT5 has previously been shown to increase the expression of serum/glucocorticoid regulated kinase 1 (SGK1) [54,73], which is known to trigger the degradation of the inhibitor protein IκBα, thus allowing nuclear translocation of the transcription factor NFκB [74]. Genes up-regulated by NFκB activation include *PTGS2* [58] and *HIF1A* [25]. Intriguingly, NFAT5 has been suggested to play a role in EMT, a process where epithelial cells adopt to a mesenchymal phenotype [31]. EMT is associated with increased invasiveness and metastatic potential of cancer cells [75]. Whether NFAT5 plays a direct role in EMT progression in endometrial cells remains to be tested.

Based on the above results, we propose that overexpression of NFAT5 might have an important role in the progression of EnCa even though the exact mechanism on enhanced NFAT5 expression corresponding with tumor aggression is unknown. There are several possible risk factors associated. EnCa is strongly associated with obesity, poor diet, and sedentary lifestyle [1,10,76]. A high-salt diet induces over-expression of inflammatory mediators, adhesive molecules, and coagulation mediators [76,77]. It was shown that a high-salt diet increased NFAT5, activating macrophages and fibrin deposition [78]. Further, excessive salt is reported to alter the expression of many pro-inflammatory cytokines such as TNF, IL-6, and PGE_2_ mediated via NFAT5 transcriptional activity [79]. 

Similarly, high glucose uptake contributes to elevated NFAT5 levels and contributes to pathophysiology, as observed earlier in diabetic retinopathy and diabetes mellitus [22,80,81]. High levels of NFAT5 are associated with the development of obesity and insulin resistance. Lee at al. reported greater than a 50-fold increase in NFAT5 expression in response to high-fat diet in mouse models [57]. It has been demonstrated that NFAT5 can epigenetically suppress the transcriptional activity of peroxisome proliferator-activated receptor gamma (PPARγ), which modulates nutrient and energy metabolism [57]. Hence, it could be postulated that in obese women, characterized expansion of adipose tissue could cause pathological conditions like hypoxia and hypothermic resistance, which mediates elevated NFAT5 expression. Thus, high dietary salt/sugar/fat consumption might upregulate NFAT5, thereby contributing to local inflammation and promoting a tumorigenic-potential-like microenvironment. 

Apart from dietary or lifestyle factors, genetic predisposition to dysregulated NFAT5 activity could also be a high risk factor. Single-nucleotide polymorphisms (SNPs) in NFAT5 introns with *cis*-expression quantitative trait loci that affect its transcriptional function are reported [14]. These SNPs are associated with the risk of high blood pressure, diabetes mellitus, diabetic nephropathy, and inflammation, suggesting that genetic variation in NFAT5 transcription might contribute to pathological phenotypes [14,22,56,82]. Further work is required to confirm these findings in relation to EnCa. 

In summary, our study indicates that high levels of NFAT5 are associated with more aggressive endometrial cancers. Further, overexpression of NFAT5 leads to activation of HIF-1α and COX2 followed by higher PGE2 levels, which may support the development of more aggressive tumors. Even though the exact underlying mechanism that drives aberrant expression of NFAT5 in tumor tissues remains elusive, the advent of selective COX2 inhibitors as anti-cancer therapies could be a useful tool for EnCa. In conclusion, NFAT5 is expressed in high-grade endometrial tumor tissue and upregulates many genes including *HIF1A* and *PTGS2*, which may participate in malignant tumor pathogenesis. 

## 4. Materials and Methods

### 4.1. Clinical Sample Collection

A series of 26 endometrial carcinomas (Table 1) from FFPE tumor samples (obtained from surgical specimens, retrospective samples from 2014–2016) from the Women’s University Hospital of Tübingen (Tübingen, Germany) were ethically obtained. All carcinomas were centrally reviewed by gynecologic pathology subspecialty pathologists to ensure that the tumor was correctly subtyped based on well-established pathological criteria. 

### 4.2. Immunohistochemistry

Immunostaining was performed on formalin fixed, paraffin-embedded archival tissue. The paraffin blocks were sliced into 2.5 μm thick sections onto glass slides. The slides were loaded onto the automated slide stainer on the VENTANA BenchMark Series Instruments (Roche Diagnostics, Mannheim, Germany) for staining. The paraffin sections were deparaffinized with Ez prep (#950-102, Roche Diagnostics). Antigen retrieval was achieved by incubating slides with CC1 (#950124, Roche Diagnostics) for 64 min at 37 °C. Endogenous peroxidases were quenched by incubating the slides with pre-primary peroxidase inhibitors. The slides were then incubated with a primary antibody for NFAT5 (#NB120-3446, Novus Bio, Wiesbaden-Nordenstadt, Germany, 1:250) for 20 min at 37 °C. The bound primary antibody was detected using an OptiView DAB IHC Detection Kit (#760-700, Roche Diagnostics) following the manufacture’s protocol. The immunohistochemical reaction was assessed with Nikon Eclipse E200 light microscope with Nikon DS-Fi1 digital camera (Nikon, Amstelveen, The Netherlands). Complete negative staining equals a score of 0, 0–10% score 1, 10–50% score 2, and >50% score 3. This analysis was independently performed by 2 pathologists (A.S., I.P.).

### 4.3. Cell Culture

Two well-differentiated endometrial carcinoma cell lines Ishikawa (type 1 endometrialcarcinoma, ECACC #99040201, Sigma-Aldrich, Taufkirchen, Germany, RRID: CVCL_2529) and HEC1a (type2 endometrial adenocarcinoma, #HTB-112, ATCC, Wesel, Germany, RRID: CVCL_0293) were cultured in a humidified atmosphere of 5% CO_2_ at 37 °C in Dulbecco’s modified Eagle’s medium F-12 (DMEM: F12) (#11039-021, Invitrogen, Darmstadt, Germany), which was phenol-free and supplemented with 10% (*v*/*v*) FBS (#10270-106, Invitrogen); the FBS was replenished every 48 h. Cells were passaged when the confluency reached at 80%. When performing experiments, cells were cultured in DMEM: F12 phenol-free supplemented with 2% (*v*/*v*) FBS medium containing antibiotics. Cells were treated with dimethyloxalylglycine (DMOG) (#D3695; Sigma-Aldrich, 0.5 mM) for 24 h. All work was carried out in a Class I laminar flow hood. Cells were routinely tested for mycoplasma and always gave a negative result.

### 4.4. Plasmid DNA Transfection

Ishikawa or HEC1a cells were plated in 6-well plates at a density of 200,000 cells per well in 2% media as described above. Ishikawa cells were transfected with plasmid DNA (2.5 ng/µL) encoding human NFAT5 in pcDNA6V5-HisC vector [54] by using Lipofectamine LTX DNA transfection reagent (#15338-100, Invitrogen) according to the manufacturer’s protocol. After 24 h the transfection mix was removed and the cells were treated with DMOG (0.5 mM) for another 24 h. Control cells remained untreated. Post treatment cells were collected for downstream analysis.

### 4.5. RNA Sequencing and Data Analysis

Total RNA samples were extracted by using miRNeasy Mini kit (#217084, QIAGEN, Hilden, Germany) according to the manufacturer’s protocol. RNA concentration and quality were measured by using a photometric measurement of nucleic acid approach with Varioskan LUX (Thermo Fisher Scientific, Sindelfingen, Germany). RNA quality was assessed with an Agilent 2100 Bioanalyzer system (Agilent Technologies, Waldbronn, Germany). Samples with high RNA integrity (>8) were selected for library construction. Using the NEBNext Ultra II Directional RNA Library Prep Kit (#E7760S, New England Biolabs GmbH, Frankfurt am Main, Germany) for Illumina and 100 ng of total RNA for each sequencing library, poly(A) selected sequencing libraries were generated according to the manufacturer’s instructions. All libraries were sequenced on the Illumina NovaSeq 6000 (Illumina, San Diego, CA, USA) platform in paired-end mode with read length 50 bp and at a depth of approx. 70 million clusters each. Library preparation and sequencing procedures were performed by the same individual and a design aimed to minimize technical batch effects was chosen. Quality of raw RNA-seq data in FASTQ files was assessed using ReadQC (https://github.com/imgag/ngs-bits) to identify potential sequencing cycles with low average quality and base distribution bias. Reads were pre-processed with skewer (version 0.2.2) and aligned using STAR (version 2.5.4a) allowing spliced read alignment to the human reference genome (GRCh37). Alignment quality was analyzed using MappingQC (https://github.com/imgag/ngs-bits) and visually inspected with Broad Integrative Genome Viewer (IGV, version 2.3.1). Based on the Ensemble genome annotation (GRCh37 v75), read counts for all genes were obtained using subread (version 1.6.0).

For differential expressed gene (DEGs) analysis, raw gene read counts were filtered to only retain genes with at least 1 cpm (count per million) in at least two samples, leaving > 15,000 genes for determining differential expression in the pair-wise comparisons be-tween experimental groups. The analysis was performed with edgeR [83,84,85] (version 3.40.2) that uses a statistical framework based on negative binomial distributions and gene-wise testing using generalized linear models. Genes with absolute log2FC ≥ 0.3, and false discover rate (FDR) < 0.05 were considered DEGs and used for downstream analysis. Functional enrichment analysis of DEGs were performed by using gprofiler2 [86] in R by using default parameter values. The gprofiler2 R package utilizes the hypergeometric test, along with correction for multiple testing, to detect statistically significant (over-represented) functional annotations from diverse set of resources such as GO [87], KEGG [88], Reactome [89], and human disease annotations [90] etc., all through a single command. In this paper, we have shown only the top 10 functional terms on the corresponding plots and have provided the complete list in the Appendix A.

### 4.6. Ingenuity Pathway Analysis (IPA)

Pathway enrichment analysis on NFAT5 overexpression in Ishikawa cells was analysied with a web-based bioinformatics application, Qiagen IPA platform. Data were analyzed with the use of QIAGEN IPA (QIAGEN Inc., https://digitalinsights.qiagen.com/IPA).

### 4.7. Messenger RNA (mRNA) Extraction and Quantitative Real-Time Reverse Transcriptase PCR (qRT-PCR)

Total RNA was extracted from cells using TRizol^TM^ reagent (#15596026, Invitrogen). One µg of total RNA was utilized to synthesize cDNA using the MaximaTM H Minus cDNA Synthesis Master Mix with dsDNase (#M1681, Invitrogen). mRNA concentration was measured by using a Nanodrop. qRT-PCR was performed on the QuantStudio 3 Real-Time PCR System (Invitrogen) by using sets of gene-specific primers (Table 2). The cycling conditions were; hold stage for 20 s at 95 °C and PCR stage with 40 cycles of 1 s at 95 °C and 20 s at 60 °C. The relative differences in PCR product amounts were quantified by the 2^−ΔΔCT^ method [91], using ribosomal L19 (*L19*) as an internal control [92]. Experiments were performed in triplicate. Melting curve was utilized to confirm amplification specificity. All the gene-specific primers used in this study was designed using Primer-BLAST (NCBI) [93] and purchased from Sigma-Aldrich. 

### 4.8. Western Blotting

Whole cell protein lysate was extracted from Ishikawa cells using Laemmli buffer as previously reported [94]. Whole cell protein lysates were heated at 95 °C for 5 min. Extracts were then loaded on to a 10% sodium dodecyl sulfate polyacrylamide gel (SDS-PAGE) using the XCell SureLock^®^ Mini-Cell apparatus (Invitrogen) followed by electrophoresis. The protein from the gel was transferred onto polyvinylidenefluoride membrane (#10600023, VWR International GmbH, Ulm, Germany). After air drying the membranes, they were activated in 100% methanol and subsequently blocked using 5% milk for 1 h at room temperature (RT). Membranes were probed overnight at 4 °C with antibodies: human NFAT5 antibody (1:1000, #NB20-3446, Novus Bio), human COX2 antibody (1:1000, #160106, Cayman Chemical, Ann Arbor, MI, USA), human pan-actin antibody (HRP-conjugate) (1:1000, #12748, Cell Signaling Technology, Frankfurt am Main, Germany) was used as loading control. After 3 × 15 min washing with 1 × TBST, the membranes were incubated with HRP-conjugated anti-rabbit secondary antibody (1:2000, #7074s, Cell Signaling Technology) at RT for 1 h. Next, after second 3 × 15 min washes, protein bands were detected using a chemiluminescent detection kit (#34580, SuperSignal™ West Pico PLUS Chemiluminescent Substrate, Thermo Fisher Scientific) and visualized by using iBright^TM^ Imaging System (Invitrogen). Bands were quantified with ImageJ Software 1.53k (National Institutes of Health, Bethesda, MD, USA) [59,94].

### 4.9. Luciferase Reporter Assay

Ishikawa or HEC1a cells were seeded onto 24-well plates at a density of 5 × 10^4^ cells/well with 10% FBS DMEM and allowed to attach for 24 h. Next, cells were transfected with HIF-1α vector (#87261, Addgene, Watertown, MA, USA) using Lipofectamine LTX with Plus reagent (#15338100, Invitrogen) as per the manufacturer’s instruction. After transfection for 24 h, cells were subjected to NFAT5 overexpression transfection followed by DMOG treatment as described above. The reporter activation was determined using the Dual-Luciferase Reporter Assay System (#E2920, Promega, Madison, WI, USA) according to the manufacturer’s instructions. 

Briefly, growth medium was removed and cells were washed with PBS. Subsequently, cells were lysed for 15 min at room temperature using 1× passive lysis buffer. Lysed cells were used for determination of luciferase activity. LAR II reagent was added to each well, and firefly luminescence was measured using a microplate reader (LUX VARIOSKAN, Thermo Fisher Scientific). Next, Stop & Glo reagent was added to each well and renilla luciferase activity was measured using a microplate reader. Three replicate wells were used for each analysis, and the results were normalized to the activity of renilla luciferase.

### 4.10. Enzyme-Linked Immunosorbent Assay (ELISA)

After transfection of Ishikawa cells with NFAT5 plasmid followed by DMOG treatment as stated above, the culture medium was harvested and stored at −80 °C. The collected cultured medium was processed for ELISA by using human prostaglandin E2 ELISA Kit (#KHL1701, Invitrogen) following the manufacturer’s instructions. The absorbance was measured with Varioskan LUX spectrophotometer (Thermo Fisher Scientific).

### 4.11. Cell-Cycle Anaylsis with Flow Cytometry

The effect of NFAT5 overexpression on Ishikawa cell cycle progression was studied with FACS approach. After treatment with NFAT5 overexpression plasmid as described above, cells and medium were collected into a 15 mL centrifuge tube and spun down at 600× *g* for 5 min. The supernatant was discarded and 1 mL of −20 °C ice-cold ethanol (#20821.330, VWR International GmbH), PBS (#D8537, Sigma-Aldrich) mixture (3:1) was added to the pellet during vortexing. The mixture was kept at −20 °C overnight, the next day washed with PBS again, 250 μL PI mix containing 50 μg/mL PI (#P4864, Sigma-Aldrich) and 100 μg/mL RNase A (#R4642, Sigma-Aldrich) were added, incubated for 30 min at 37 °C, and subjected to flow cytometry (BD LSRFortessa™ Cell Analyzer, BD Biosciences, Heidelberg, Germany) for cell cycle analysis. The data were analyzed by FlowJo^TM^ software 10.8.1 (FlowJo, Ashland, OR, USA). 

### 4.12. BrdU ELISA Cell Profileration Assay

The effect of NFAT5 overexpression on Ishikawa proliferation was measured using BrdU cell proliferation assay (#QIA58, Sigma-Aldrich). Post treatment with NFAT5 overexpression plasmid as described above, the cells were immunolabelled for BrdU and the cells incubated for an additional 24 h. Incorporated BrdU was detected by the BrdU cell proliferation assay as instructed in the manufacture protocol. Fluorescence was measured using a microplate reader (LUX VARIOSKAN, Thermo Fisher Scientific).

### 4.13. Wound Scratch Assay

Ishikawa/HEC1a cells were seeded in six-well plates at a concentration of 200 × 10^3^ cells per well. After reaching 100% confluency, cells were deprived of serum for 24 h and scratched with a sterile P200 pipette tip (#613-1096, VWR International GmbH) as previously described [59]. After removal of the debris by repeated washes, cells were subjected to respective treatment (control/NFAT5 overexpression) and scratch wound closure was closely monitored by microscopy (EVOS M7000 cell imaging system, Thermo Fisher Scientific) capturing bright field images of the same field with a 4× objective at 0 h and 24 h. The percentage of wound are closure was calculated with ImageJ software.

### 4.14. Statistics

The data are given as arithmetic mean ± SEM, the number of independent biological experiments were denoted as *n*. The data were analyzed for significance using un-paired Student’s *t*-test using GraphPad Prism (GraphPad Software 7.0, San Diego, CA, USA). Statistical significance was considered when *p* value was less than < 0.05. the clinical table was analyzed using Stata statistical software version 17.2 (StataCorp LLC, College Station, TX, USA). We compared categorical variables using χ^2^ tests.

## Figures and Tables

**Figure 1 ijms-25-03666-f001:**
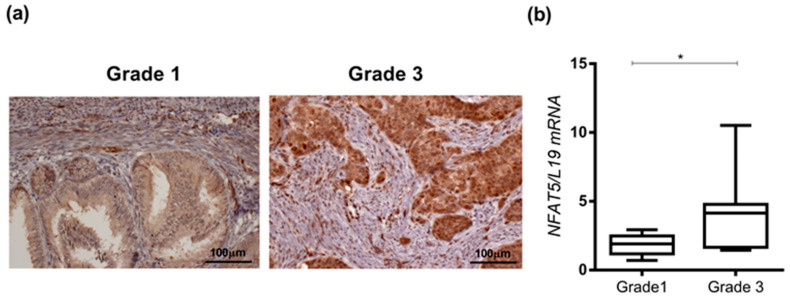
NFAT5 expression in Grades 1 and 3 endometrial cancer tissues. (**a**). Representative images of NFAT5 immunohistochemistry expression analysis in Grade 1 (*n* = 15) and Grade 3 (*n* = 11) endometrial cancer tissue. Staining shows excessive NFAT5 expression in Grade 3 tumor samples compared to Grade 1. Scale bar—100 µm. (**b**). In parallel, mRNA expression level of *NFAT5* from FFPE tissue samples was quantified by qRT-PCR, *, *p* < 0.05 based on unpaired *t*-test. Data were normalized to ribosomal housekeeping gene, *L19*.

**Figure 2 ijms-25-03666-f002:**
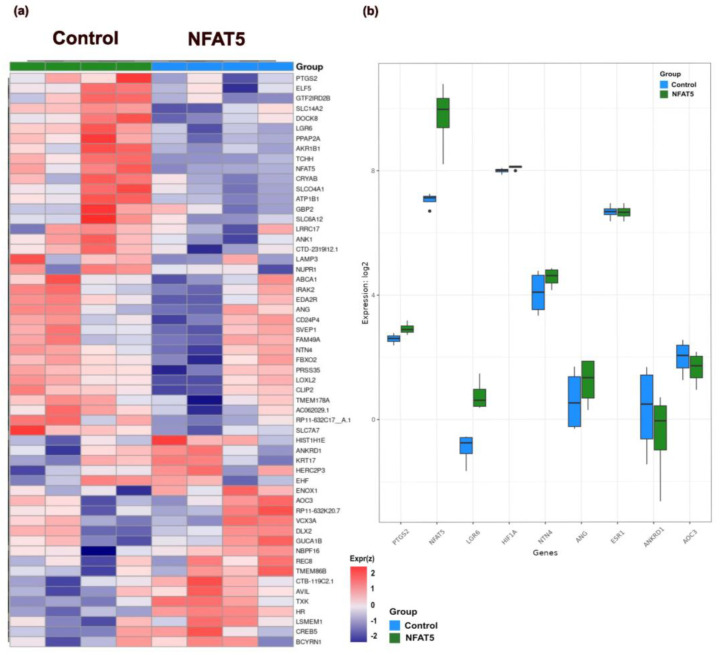
Gene expression alteration in Ishikawa cells with NFAT5 overexpression. (**a**). Heat-map shows gene expression alteration by NFAT5 overexpression in Ishikawa cells (FDR < 0.05 and log2FC ≥ 0.3). Upregulation and downregulation of genes are shown by red and blue color coding, respectively. (**b**). Box and whisker plots of log2 fold change of genes of interest in control and NFAT5 overexpressed Ishikawa cells.

**Figure 3 ijms-25-03666-f003:**
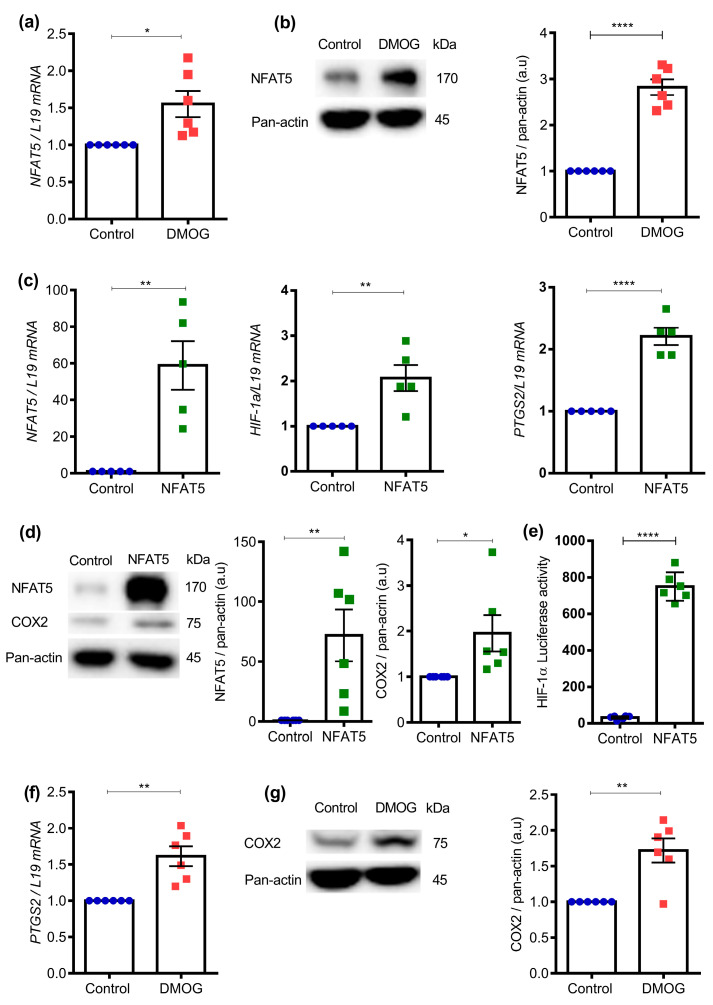
Effect of NFAT5 on COX2 transcript and protein levels in Ishikawa cells. (**a**). Ishikawa cells were treated with 0.5 mM DMOG for 24 h. mRNA expression level of *NFAT5* was quantified by qRT-PCR. Data were normalized to *L19* and presented as mean ± SEM. (*n* = 6; *, *p* < 0.05). (**b**). NFAT5 protein abundance was investigated by SDS-PAGE and western blot analysis. Ishikawa cells were treated with 0.5 mM DMOG for 24 h. Data were normalized to each corresponding level of pan-actin and shown as mean ± SEM. (*n* = 6; ****, *p* < 0.0001, a.u: arbitrary unit). (**c**). mRNA expression level of *NFAT5*, *HIF1A*, and *PTGS2* were quantified by qRT-PCR. Ishikawa cells were transfected with NFAT5 overexpression plasmid for 24 h (*n* = 5; **, *p* < 0.01, ****, *p* < 0.0001). (**d**). NFAT5 and PTGS2 protein abundance were investigated by SDS-PAGE and western blot analysis using the indicated antibodies. Ishikawa cells were transfected with NFAT5 overexpression plasmid for 24 h (*n* = 6; *, *p* < 0.05; **, *p* < 0.01). (**e**). Luciferase activity of HIF-1α that was normalized to renilla post Ishikawa cells transfected with NFAT5 overexpression plasmid for 24 h (*n* = 6; ****, *p* < 0.0001). (**f**). mRNA expression level of *PTGS2* was quantified by qRT-PCR. Ishikawa cells were treated with 0.5 mM DMOG for 24 h (*n* = 6; **, *p* < 0.01). (**g**). COX2 protein abundance was investigated by SDS-PAGE and western blot analysis using the indicated antibodies. Ishikawa cells were treated with 0.5 mM DMOG for 24 h (*n* = 6; **, *p* < 0.01).

**Figure 4 ijms-25-03666-f004:**
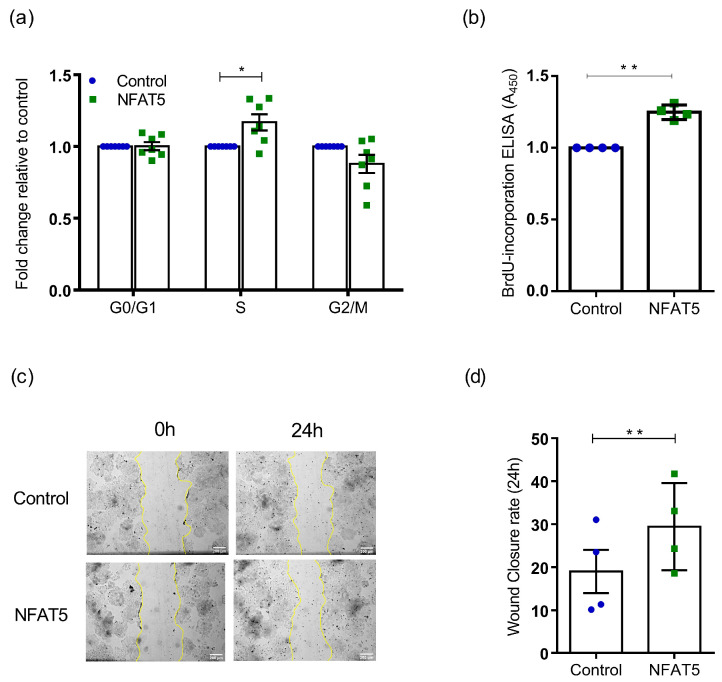
Effect of NFAT5 overexpression in Ishikawa biological activity. (**a**). FACS assisted cell cycle analysis on control and NFAT5 overexpressed Ishikawa cells (*n* = 7; *, *p* < 0.05). (**b**). Cell proliferation analysis on control and NFAT5 overexpressed Ishikawa cells with BrdU ELISA assay (*n* = 4; **, *p* < 0.01). Data were normalized to each control and shown as mean ± SEM. (**c**). Representative bright field images of wound healing scratch assay on the control and NFAT5-overexpressed Ishikawa cells. (**d**). Wound closure rate on the control and NFAT5 overexpressed Ishikawa cells 24 h post scratch (*n* = 4; **, *p* < 0.01), scale bar—200 µm.

**Figure 5 ijms-25-03666-f005:**
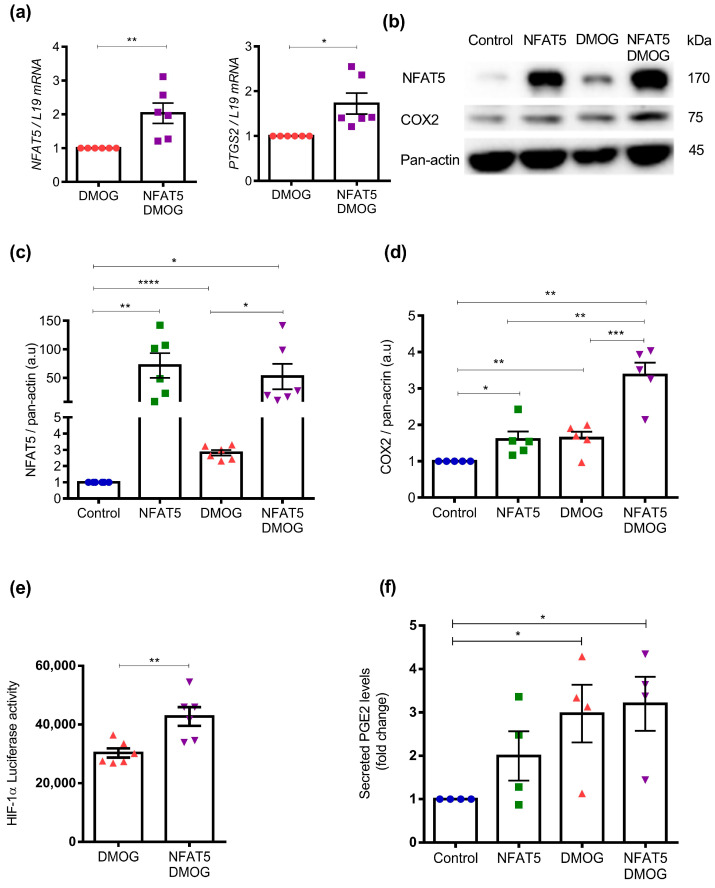
Synergistic effect of NFAT5 overexpression and DMOG on COX2 transcript and protein levels in Ishikawa cells. (**a**). mRNA expression level of *NFAT5* and *PTGS2* quantified by qRT-PCR. Ishikawa cells were treated with 0.5 mM DMOG for 24 h post with or without 24 h transfection with NFAT5 overexpression plasmid. Data were normalized to L19 and presented as mean ± SEM. (NFAT5, *n* = 6; *, *p* < 0.05; **, *p* < 0.01). (**b**–**d**). NFAT5 and COX2 protein abundance were investigated by SDS-PAGE and western blot analysis using the indicated antibodies. Ishikawa cells were subjected to with or without NFAT5 transfection followed by treatment with or without DMOG. Data were normalized to each corresponding level of pan-actin and shown as mean ± SEM. (*n* = 5; *, *p* < 0.05; **, *p* < 0.01; ***, *p* < 0.001, ****, *p* < 0.0001, a.u: arbitrary unit). (**e**). Effect of hypoxia on HIF-1α induction with or without NFAT5 transfection using Luciferase promoter assay. Data shown as mean ± SEM. (*n* = 6; **, *p* < 0.01). (**f**). Effect on PGE2 levels with or without NFAT5 transfection, followed by treatment with or without DMOG (0.5 mM, 24 h). Data shown as mean ± SEM. (*n* = 4; *, *p* < 0.05).

**Table 1 ijms-25-03666-t001:** Clinical characteristics of study cohort.

Clinical Characteristics	Total	NFAT5 Score			*p*-Value
		1	2	3	
Age > 60 years	15 (100%)	5(33.3%)	6 (40%)	4 (26.7%)	*p* = 0.941
Age < 60 years	11 (100%)	3 (27.3%)	5 (45.5%)	3 (27.3%)	
Premenopausal	7 (100%)	3 (42.9%)	3 (42.9%)	1 (14.3%)	
Postmenopausal	19 (100%)	5 (26.3%)	8 (42.1%)	6 (31.6%)	*p* = 0.599
Endometriod	24 (100%)	8 (33.3%)	9 (37.5%)	7 (29.2%)	*p* = 0.228
Serous	2 (100%)	0 (0%)	2 (100%)	0 (0%)	
Grade 1/2	15 (100%)	8 (53.3%)	7 (46.7%)	0 (0%)	*p* < 0.001
Grade 3	11 (100%)	0 (0%)	4 (36.4%)	7 (63.6%)	
pT1a	15 (100%)	7 (46.7%)	7 (46.7%)	1 (6.7%)	*p* = 0.091
pT1b	6 (100%)	0 (0%)	3 (50%)	3 (50%)	
pT2	2 (100%)	1 (50%)	0 (0%)	1 (50%)	
pT3a	3 (100%)	0 (0%)	1 (33.3%)	2 (66.7%)	
pT1a	15 (100%)	7 (46.7%)	7 (46.7%)	1 (6.7%)	*p* = 0.014
>pT1b	11 (100%)	1 (9.1%)	4 (36.4%)	6 (54.5%)	
Regional Nodes pN0	21 (100%)	7 (33.3%)	10 (47.6%)	4 (19%)	*p* = 0.176
Regional Nodes pN1	5 (100%)	1 (20%)	1 (20%)	3 (60%)	
Metastatsis 0	20 (100%)	7 (35%)	10 (50%)	3 (15%)	*p* = 0.043
Metastatsis 1	6 (100%)	1 (16.7%)	1 (16.7%)	4 (66.7%)	
Lymph Vessel L0	19 (100%)	8 (42.1%)	7 (36.8%)	4 (21.1%)	*p* = 0.114
Lymph Vessel L1	7 (100%)	0 (0%)	4 (57.1%)	3 (42.9%)	

**Table 2 ijms-25-03666-t002:** List of the human primer sequences used in the study.

Gene	Primer Sequence
*L19*	Forward (5′-3′): GCGGAAGGGTACAGCCAA
	Reverse (5′-3′): GCAGCCGGCGCAAA
*NFAT5*	Forward (5′-3′): GAGCAGAGCTGCAGTAT
	Reverse (5′-3′): AGCTGAGAAAGCACATAG
*PTGS2*	Forward (5′-3′): GCTCAAACATGATGTTTGCATTC
	Reverse (5′-3′): GCTGGCCCTCGCTTATGA
*HIF1A*	Forward (5′-3′): TCTGGACTTGCCTTTCCTTCTC
	Reverse (5′-3′): AACTTATCTTTTTCTTGTCGTTCGC

## Data Availability

Expression data have been submitted to the Gene Expression Omnibus (GEO) repository with accession number GSE134319. The remaining data presented in this study are available upon reasonable request from the corresponding author.

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
