# Peer review of "Rel Family Transcription Factor NFAT5 Upregulates COX2 via HIF-1α Activity in Ishikawa and HEC1a Cells"

_ijms, 2024, doi:10.3390/ijms25073666_

Round 1

Reviewer 1 Report

Comments and Suggestions for Authors

The authors of this paper showed upregulation of NFAT5 expression in higher grade endometrial cancer samples. They then used the Ishikawa cell line to show that NFAT5 overexpression leads to increased HIF-1α and COX2 expression. Interestingly, DMOG treatment induced increased NFAT5, suggesting synergistic interaction between NFAT5 and hypoxia.  There are three major issues: 1. The authors claimed that “NFAT5 and HIF-1α mutually up-regulate each other”, which is not entirely clear. They showed that NFAT5 overexpression increases HIF-1α, but did not show HIF-1α overexpression cause increase in NFAT5.  2. Related to 1, the authors claimed that “the mutual stimulation of HIF-1α and NFAT5 thus establishes a positive feedback loop augmenting the response to hypoxia” based on the following result:  DMOG treatment (HIF-1α stabilization) -> NFAT5 increase -> HIF-1α transcript increase. The authors seem to equate HIF-1α stabilization with HIF-1α transcript increase, which may not be the same thing. 3. In the Discussion, the authors claim “Further, over expression of NFAT5 leads to activation HIF-1α, COX2 and higher PGE2 levels which may support the development of more aggressive tumours”. However, the results in Fig. 4f reveals there is no increase in PGE2 with NFAT5 overexpression alone and only slightly increased PGE2 with NFAT5 overexpression combined with DMOG treatment.

Reviewer 2 Report

Comments and Suggestions for Authors

In this study, the authors evaluated the expression significance of NFAT5 in endometrial cancer patients and the function using a cancer cell line. Interestingly, the NFAT5 expression was upregulated in high-grade endometrial cancer. Moreover, the in vitro analysis clarified that NFAT5 was associated with the HIF1A and COX2 regarding cancer aggressiveness. These data were significant and informative; however, the data might be insufficient. Therefore, I raised several points to improve the content of the report. 

Query

1. Method of the IHC should be revised.

2. For the functional analysis of NFAT5, the authors used only one cell line. At least two cell lines should be used to validate the NFAT5 function.

3. How about the tumor-specific expression of NFAT5 compared to the non-cancerous part?

4. Typo in the result section should be checked: 

"Ishikawa cells (a well-used model of adenocarcinoma cancer)"

5. The relation of NFAT5, HIF1A, and COX2 should be checked in clinical endometrial tumor samples using multi-color IHC. Clinical data showed only the relation between NFAT5 and grading. Currently, there is not enough data available to support the conclusion.

6. The manuscript should describe the primer sequence because nobody can judge whether the PCR experiments are reliable without the data. 

7. L19 might not be a widespread internal control gene. However, some researchers suggested that ribosomal RNA is an excellent internal gene in endometrial cancer. The authors should cite the paper regarding L19 as a reference gene in the method section.

8. The papers regarding the WB methods should be cited, as well as other references. Moreover, all antibody product numbers should be rechecked. When did the authors use the GAPDH antibody for WB?

Reviewer 3 Report

Comments and Suggestions for Authors

In the submitted manuscript authors studied the role of transcription factor NFAT5 in the regulation of gene expression in endometrial cancer.

Although relatively robust, this manuscript suffers from lots of ambiguities and imprecisions, while sometimes it is very sloppy and superficial. Also, some important data were not provided, because of which there is an impression that authors were either cherry picking or hiding the results!

1) 'Abstract' is a little bit too scarce since lots of important information was not mentioned (e.g., RNA-Seq).

2) In 'Introduction', to much data were provided for NFAT5, while too little for molecular mechanisms behind endometrial cancer development and progression.

3) At many parts in the text, it is unclear weather authors meant gene (mRNA) or protein. Therefore, authors should carefully inspect https://www.genenames.org/ and https://www.uniprot.org/uniprotkb/ and solely use only approved gene/protein names and symbols (e.g., PTGS2 instead of COX2, while definitively not Greek letters) and uniformly differentiate style between gene (in italic) and protein symbols.

Also, there is no need to capitalize first letters of full gene/protein names, as well as chemical compounds (like dimethyloxalylglycine).

4) Supplementary tables were not provided for reviewing!

5) Fig S2 is in redundant and in essence meaningless without detail description, especially since it comes from the commercial software!

6) For all used software and packages, especially those used for RNA-seq analysis, original references must be cited.

7) Log2FC values should be provided also for those significantly upregulated genes (page 3).

8) Since authors haven't provided cumulative results of NFAT5 protein expression analysis, especially results of statistical analysis, what is inexplicable, Fig S1 is meaningless! What specifically rises suspicion is the statement "A total of 26 samples were investigated and were representative." (page 2), what could be interpreted that authors intentionally chose only those samples which fit their presumption that NFAT5 has higher expression only in high grade (G3) samples! Therefore, cumulative or per sample results of IHC staining together with patients' clinico-pathological characteristics MUST be provided, especially since only 26 tumor tissue samples in essence present a small sample size!

9) All abbreviations should be explained after first mentioning in the main text (e.g., FFPE, GO, IPA, HRE, FDR, ELISA).

10) It is unclear how Figure 2 was created and what it actually presented on it, since Figure 2 legend is too uninformative. E.g., it is unclear which are those genes presented in the heat map (a), since for instance it was written that 369 DEGs were found; what are those therms and why only 6 of them were presented in panel (b), etc.

11) Actual p-values must be provided in the abstract and main text whenever the results of statistical analyses (comparisons) were described, just asterisks on graphs are not enough!

12) All graphs should be presented with mean +- standard deviation, not SEM, because presenting SEM instead of SD in your case could be seen as a form of cheating!

13) For the sake of reproducibility (and credibility), both primer sequences and qPCR cycling conditions must be provided, especially since the mRNA expression of only three genes was presented! Also, proper notion of formula used for relative mRNA expression calculation is "2-ΔΔCt", while its reference PMID: 11846609 should be cited.

14) All used instruments must be mentioned (e.g., platereader used for ELISA), and their company and its country of origin properly stated.

15) Authors should carefully check text that all used methods were described with enough details so anyone could repeat them. For instance, luciferase reporter assays and Ingenuity Pathway Analysis (IPA) were not mentioned in '4. Materials and Methods'!

16) Maybe it is true that Ishikawa cells (BTW, their Resource Identification Initiative ID "(RRID:CVCL_2529)" should be provided in "Cell Culture" section) are a well-used model of endometrial adenocarcinoma, however, there is not a single proof that all observed results would be exactly the same in any other endometrial adenocarcinoma cell line! Therefore, this should be reflected in the title of this manuscript, because otherwise it could be misleading.

17) Authors maybe discovered, to a certain point, how NFAT5 regulates expression of its target genes, but the repercussions of NFAT5 overexpression on the biology of endometrial cancer cells were unfortunately not studied, while this would be even more important and interesting! Therefore, authors should conduct some, at least in vitro, cell-based assays to inspect impact of NFAT5 overexpression on Ishikawa cells biology (proliferation, migration, etc.).

Comments on the Quality of English Language

There are lots of unclear or wrong phrases:

Page 3/Line 113: I believe the first phrase "the effect of," is redundant.

P4/L134: Proper is "gene expression alteration".

P4/L142: Phrase "transfection with NFAT5" is ambiguous since you didn't use protein.

P4/L146: It is unclear what means "Sup 2".

P5/L170: Statement "To test this hypothesis in endometrial cells" is wrong since you haven't used endometrial cells, but endometrial carcinoma cells!

P6/L189: Rennila's luciferase is definitively not a "housekeeping"!!!

P9/L313: Correct would be "upregulates many genes".

P10/L330: Put "at" instead of "@".

P10/L337-338: Statement "This analysis was independently performed by 2 reviewers (A.S., I.P)" is awkward since it is unclear what "reviewers" have to do with IHC?!

P10/L367: ">8" should be put under the parentheses.

P11/L403: Actual name of that tool is Primer-BLAST, while its reference PMID: 22708584 should also be cited!

P11/L408 and 423: The references were improperly cited.

Figures 3 and 4: It is unclear what means "(a.u.)" on graphs presenting pan-actin protein expression.

There are lots of redundant phrases or words repetition:

Figure 1 legend: "Scale bar 100μm." and "and presented as box and whiskers" are redundant.

P3/L122-123: Phrase ", in Ishikawa cells" is redundant.

Figure 3 and 4 legends should be shortened with all conditions which were repeating (e.g., "was quantified by qRT-PCR", "Data were normalized to L19 and presented as mean ± SEM", etc.) should be mentioned only once, at the beginning.

Round 2

Reviewer 1 Report

Comments and Suggestions for Authors

Authors have addressed all concerns.

Author Response

Thank you

Reviewer 2 Report

Comments and Suggestions for Authors

These data were significant and informative; however, the impact of data might be insufficient. 

Author Response

Thank you

Reviewer 3 Report

Comments and Suggestions for Authors

Authors have substantially improved quality of this manuscript through revision, however, there are still few things which must be improved before this manuscript is suitable for publication:

1) In Table 1 there should be provided p-values for all statistical analyses (e.g., age, nodes, etc.), not just those significant ones.

2) Figure 2 is too small and thus unreadable. Panel a and b should be put horizontal and enlarged.

3) qPCR cycling conditions must be provided in "Messenger RNA (mRNA) extraction and Quantitative Real-time reverse transcriptase PCR (qRT-PCR)" section, while HIF1A is approved symbol of hypoxia inducible factor 1 subunit alpha gene.

4) IPA should be properly cited: https://digitalinsights.qiagen.com/citation-guidelines/
